# Advancements in Optical Resonator Stability: Principles, Technologies, and Applications

**DOI:** 10.3390/s24196473

**Published:** 2024-10-08

**Authors:** Huiping Li, Ding Li, Qixin Lou, Chao Liu, Tian Lan, Xudong Yu

**Affiliations:** 1College of Advanced Interdisciplinary Studies, National University of Defense Technology, Changsha 410073, China; 13938115926@163.com (H.L.); liding97@nudt.edu.cn (D.L.); louqix@163.com (Q.L.); liuchao0622@126.com (C.L.); a1476073873@163.com (T.L.); 2Nanhu Laser Laboratory, National University of Defense Technology, Changsha 410073, China

**Keywords:** optical resonator, optical resonator stability, vibration sensitivity, mechanical vibration isolation, thermal noise control, temperature control

## Abstract

This paper provides an overview of the study of optical resonant cavity stability, focusing on the relevant principles, key technological advances, and applications of optical resonant cavities in a variety of high-precision measurement techniques and modern science and technology. Firstly, the vibration characteristics, thermal noise, and temperature characteristics of the reference cavity are presented. Subsequently, the report extensively discusses the advances in key technologies such as mechanical vibration isolation, thermal noise control, and resistance to temperature fluctuations. These advances not only contribute to the development of theory but also provide innovative solutions for practical applications. Typical applications of optical cavities in areas such as laser gyroscopes, high-precision measurements, and gravitational wave detection are also discussed. Future research directions are envisioned, emphasising the importance of novel material applications, advanced vibration isolation technologies, intelligent temperature control systems, multifunctional integrated optical resonator design, and deepening theoretical models and numerical simulations.

## 1. Introduction

A photonic feedback mechanism within an optical enclosure is usually formed using two planar or inward-curved mirrors positioned at right angles to the medium’s operational axis [1]. This platform enables the enhancement of light signals when population inversion is achieved. Through emphasizing the enhancement of light of particular frequencies and angles, and concurrently inhibiting other frequencies and directions, optimal amplification is guaranteed. Photons that veer away from the cavity’s central axis swiftly evade, reducing their interaction with the operational substance. Conversely, photons aligned along the axis oscillate inside the chamber, bouncing off the mirrors repeatedly and interacting with stimulated particles, which induces stimulated emission. This procedure leads to the multiplication of organized photons, generating a robust, one-way beam of light characterized by consistent frequency and phase—known as the laser. To remove the laser from the enclosure, a single mirror possesses partial reflectivity. This enables some of the light to pass through as usable laser emission, while the rest is reflected to sustain photon amplification inside the enclosure.

As an important part of the laser, the optical resonator primarily provides optical feedback to the system. Because of its ability to select the frequency of the laser mode, the optical resonator is sometimes used for spectral analysis. In addition, especially in the case of a super-stabilised cavity, it plays a crucial role in laser frequency stabilisation. Typically, the laser output frequency is susceptible to changes over time due to environmental disturbances and intrinsic instabilities, which manifest themselves as short-term jitter and long-term drift. Frequency stabilisation techniques, including passive and active stabilisation, are used to ensure the stability of the laser frequency. The essence of these techniques is to maintain the stability of the optical resonator length within the laser. Passive stabilisation [2] mainly consists of thermostatic control, vacuum-sealed enclosures, and vibration isolation to mitigate the effects of ambient temperature and mechanical vibration on frequency stability. Given the high sensitivity of laser frequencies to temperature and vibration variations, passive stabilisation cannot guarantee long-term, high-order frequency stability despite comprehensive interference mitigation measures. This limitation affects the accuracy of precise absolute distance measurements.

Therefore, in order to improve the stability of laser frequency and realize high stability and ultra-narrow linewidth output, active frequency stabilization technology has been developed [3]. These techniques start with choosing a highly stable reference frequency. When the laser frequency deviates from the reference due to environmental changes, an error signal deviation is generated by detecting the frequency. Subsequently, the servo feedback control system adjusts the length of the optical resonator to stabilize the laser frequency at the reference point to achieve stable laser output. Pound–Drever–Hall (PDH) laser frequency stabilization technology [4], also known as phase modulation heterodyne technology, uses Fabry–Perot (F-P) interferometer as reference frequency, which represents an active stabilization form. Because of its strong anti-interference ability, high stability, fast servo response, and low possibility of losing lock, it has become a widely used frequency stabilization method. In 1983, Drever R et al. [5] successfully locked the frequency of the gas laser to the resonant frequency of a Fabry–Perot interferometer in an optical reference cavity and realized a laser frequency with a line width of less than 100 Hz and stable dye laser. Since then, PDH stabilization technology has been named after Pound R V and Drever R. The change in cavity length directly affects the frequency stability of laser, so controlling the stability of optical resonator has become a research hotspot in this field.

This paper makes significant contributions to the study of optical resonant cavity stability by systematically reviewing relevant principles, key technological advances, and their applications in high-precision measurement techniques and modern science. Firstly, the paper provides a detailed analysis of the vibration characteristics, thermal noise, and temperature characteristics of the reference cavity, laying a foundation for understanding the performance of resonant cavities. Subsequently, it focuses on the advances in key technologies such as mechanical vibration isolation, thermal noise control, and resistance to temperature fluctuations. These advancements not only enrich theoretical research but also offer innovative solutions for practical applications. Furthermore, the paper discusses the importance of optical cavities in typical applications such as laser gyroscopes, high-precision measurements, and gravitational wave detection, showcasing their broad application potential. Future research directions emphasize the significance of novel material applications, advanced vibration isolation technologies, intelligent temperature control systems, and multifunctional integrated optical resonator designs while advocating for the deepening of theoretical models and numerical simulations. These contributions not only promote the development of the field of optical resonant cavities but also provide theoretical support and practical evidence for the advancement of related technologies.

## 2. Correlation Principle

### 2.1. Vibration Characteristics of Reference Cavity

Although an ultra-stable laser with very-high-frequency stability can be generated by using an optical reference cavity as a frequency reference, ground vibration and acoustic noise will interfere with the length stability of the reference cavity, thus affecting the stability of the ultra-stable laser. When the external vibration frequency closely matches or coincides with the natural frequency of the reference cavity, this effect is particularly obvious, resulting in more obvious interference and a significant decline in the frequency stability of the laser.

The ground vibration is transmitted to the cavity through the supporting structure, which leads to the elastic deformation of the material constituting the optical resonator. This deformation accelerates the relative displacement and inclination of the two mirrors describing the resonant cavity, which eventually leads to the change in the optical path length of the resonant cavity. As shown in Figure 1, a cylindrical cavity of length *L* is depicted, positioned vertically with a padded plate, experiencing vertical compression due to gravity. At this moment, the interaction between different directions caused by the Poisson ratio being non-zero is disregarded. The cavity’s length alteration is
(1)ΔL/L=−ρgL/2E,
where *g* is the gravitational acceleration, and *p* and *E* are the mass density and the elastic modulus of the cavity material, respectively. For a 10 cm cavity made from ULE, the change in the optical length due to gravity is $1.6\times 10^{-8}$ (fractional), resulting in a frequency change of ∼10 MHz/g at 532 nm.

Therefore, this vibration directly causes frequency noise in the laser locked on the optical resonator. At present, there are usually two methods to isolate vibration: active and passive vibration reduction. Passive vibration reduction adopts spring structure or cushioning material [7], and the active method uses displacement actuators (such as piezoelectric ceramics and electromagnetic coils) to control the movement of the platform where the optical resonator is located to offset external vibration [8,9,10]. These two vibration isolation methods can filter out most of the high-frequency signals of vibration, but it is difficult to attenuate the low-frequency vibration. In addition, this high-precision passive or active control system is costly and difficult to reduce in size.

In addition to implementing different vibration isolation strategies, reducing the inherent vibration sensitivity of the optical resonator itself is also an important method [11]. By optimizing the geometric configuration, supporting structure, and positioning of the optical resonator, the sensitivity of the cavity length to vibration stimulation can be significantly reduced.

### 2.2. Thermal Noise of Reference Cavity

According to the third law of thermodynamics, nothing can reach absolute zero, and the internal molecules of each reference cavity have been in an irregular state of thermal motion, which is macroscopically manifested as the cavity length fluctuation of the reference cavity. According to the fluctuation–dissipation theory (FDT) [12], the expression of the fluctuation spectral density of the system can be obtained as follows [13]:(2)Sx(f)=kBTπ2f2Re[Y(f)],
where *S_x_*(*f*) represents the double-sideband power spectral density of displacement, *k_B_* is Boltzmann constant, *T* is thermodynamic temperature (unit is *K*), *f* is Fourier frequency, and mechanical impedance function *Y*(*f*) = *2πifx*(*f*)/*F*(*f*) represents the Fourier transformation of *x*(*f*) relative to the admittance of an external force *F*(*f*) acting on a certain material. The length stability of a reference cavity, which determines the frequency stability of a cavity-stabilized laser, is influenced by the fluctuation effects arising from the cavity itself, mirror substrates, and mirror coatings. This phenomenon is termed the thermal noise limit of the reference cavity. The thermal noise in the reference cavity primarily comprises Brownian noise [14,15], thermoelastic noise [16,17], and thermal refraction noise [18], with Brownian noise [19] exhibiting the largest amplitude, several orders of magnitude higher than the other two. In this context, only Brownian noise is elaborated upon.

According to the displacement spectral density of the reference cavity, Numata et al. [20] pointed out the influence of Boulanger noise on the length stability of the reference cavity. The displacement spectral density of the whole reference cavity consists of cavity displacement spectral density $S_{x}^{sp}(f)$, two cavity mirror substrate displacement spectral densitie $S_{x}^{sb}\left(f\right)$, and two cavity mirror coating displacement spectral densities $S_{x}^{ct}\left(f\right)$, which can be expressed as
(3)$$\begin{array}{ll} S_{x}(f) & =\frac{4k_{B}T}{\pi fF_{0}^{2}}(U_{sp}\phi_{sp}+2U_{sb}\phi_{sb}+2U_{ct}\phi_{ct})\\ & =S_{x}^{sp}(f)+2S_{x}^{sb}(f)+2S_{x}^{ct}(f) \end{array},$$
where *U_sp_*, *U_sb_*, *U_ct_*, *φ_sp_* and *φ_sb_*, and *φ_ct_* are the maximum elastic energy and mechanical loss angle of a reference cavity, cavity mirror substrate, and cavity mirror coating, respectively. Given that the diameter of the laser cavity mirror vastly exceeds the waist size of the laser beam, the spectral density of displacement of the cavity mirror substrate can be defined as follows [21]:(4)$$S_{x}^{sb}(f)=\frac{4k_{B}T}{\pi f}\frac{1-\sigma^{2}}{2\sqrt{\pi }E\omega }\phi_{sb},$$
where *E* is the elastic modulus of the cavity material, *ω* is the waist radius of the laser light field, and *σ* is the Poisson’s ratio of the material. Cavity mirror coating can be regarded as a thin layer with the thickness of *d_ct_* [22,23] on the front surface of the substrate. Assuming that its mechanical loss angle is also uniform, the displacement spectral density of the coating is expressed as
(5)$$S_{x}^{ct}(f)=S_{x}^{sb}(f)\frac{2}{\sqrt{\pi }}\frac{1-2\sigma }{1-\sigma }\frac{\phi_{ct}}{\phi_{sb}}\frac{d_{ct}}{\omega },$$

Assuming that the outer diameter of the cavity is *R_sp_*, the inner diameter is *r_sp_* and the cavity length is *L*, the displacement power spectral density of the cavity can be expressed as [24]
(6)$$S_{x}^{sp}(f)=\frac{4k_{B}T}{\pi f}\frac{L}{2\pi E(R_{sp}^{2}-r_{sp}^{2})}\phi_{sp},$$

The power spectral density of the displacement (length) of the reference cavity can be converted into the frequency noise power spectral density of the laser according to *S_y_*(*f*) = *S_x_*(*f*)/*L*^2^. Because the noise of the cavity length changes with 1/*f*, it belongs to frequency flicker noise, so the thermal noise of the reference cavity corresponds to a fixed frequency stability, which is expressed by Allen variance as follows:(7)$$\sigma_{y}=\sqrt{2\ln(2)S_{y}(f)f},$$

Commonly used reference cavity materials at room temperature are ultra-low-thermal-expansion-coefficient glass (ULE), cavity mirror materials ULE or fused silica (FS), and cavity mirror coating materials are generally Ti_2_O_5_/SiO_2_. According to the calculation of various material properties, the thermal noise of coating is the largest among the three parts of the reference cavity. According to Equations (4)–(6), the thermal noise limit of the reference cavity can be reduced by increasing the length of the reference cavity, selecting a cavity mirror with a larger radius of curvature, and selecting a cavity mirror substrate with smaller mechanical loss [25], so that the reference cavity can work in a low-temperature environment, thus improving the frequency stability limit that the cavity-stabilized laser can achieve.

### 2.3. Temperature Characteristics of Reference Cavity

The inevitable fluctuation of ambient temperature will cause a change in the length of the reference cavity, thus reducing its length stability and further affecting the frequency stability of the laser. Therefore, materials with low thermal expansion coefficients are usually selected to construct the reference cavity and the cavity mirror substrate. Examples of such materials include the ultra-low coefficient of thermal expansion glass such as ULE, glass ceramics such as Zerodur, and monocrystalline silicon.

ULE glass is usually used as the cavity material for ultra-stable optical reference cavities working at room temperature. The zero expansion temperature point of ULE materials is usually close to room temperature. However, accurate testing is needed to accurately determine the temperature. Generally, a ULE material or FS is selected as the material of the cavity mirror, which provides lower thermal noise, although its thermal expansion coefficient is about two orders of magnitude larger than the ULE material. In order to solve the thermal sensitivity problem related to the FS cavity mirror, adding a ULE compensation ring around the cavity mirror has proved to be an effective solution [26].

In practical applications, it is desirable to keep the zero thermal expansion temperature slightly higher than the ambient temperature to reduce the challenge of temperature control. The traditional “sandwich” structure is a cost-effective solution. In this structure, the thermal expansion of the cavity is reduced by placing ULE rings around the mirror. However, it provides a narrow temperature tuning range of several Kelvin, which is usually not enough to adjust the zero thermal expansion temperature close to the expected value [26]. Alternatively, the improved “reentrant” configuration combined with the FS compensation ring allows a wide adjustment range of zero thermal expansion temperature. However, this configuration is most suitable for long cavity design because it is bulky and sensitive to extrusion force, which brings challenges to miniaturization cavity design [27].

Due to the challenge of accurately determining the zero expansion temperature of the reference cavity and the limitation of the accuracy of active temperature control, it is an effective method to reduce the sensitivity of the reference cavity to external temperature changes by adopting passive heat insulation measures in the vacuum cavity and improving the heat transfer resistance from the external environment to the reference cavity [28,29].

## 3. Research Progress of Key Technologies

In the study of optical resonator stability, the progress of key technologies not only promotes the in-depth development of theory but also provides innovative solutions for practical applications. With the continuous improvement of the performance requirements of optical resonators, researchers are committed to solving the stability problems caused by mechanical vibration, thermal noise, and temperature fluctuation [8,9,10,20]. Overcoming these challenges depends on a series of innovative technologies, including efficient vibration isolation methods, effective suppression of thermal noise, and advanced temperature control systems.

### 3.1. Mechanical Vibration Isolation

Reducing the frequency noise of optical resonators caused by vibration is a key step to developing highly stable optical resonators. In recent years, various research groups have used finite element methods to simulate the micro-deformation of optical reference cavities under different vibration conditions. Then, they analyzed how different cavity geometries and installation techniques affect cavity deformation.

In 2006, Nazarova et al. [8] used finite element analysis to reduce the vibration sensitivity of the optical reference cavity. Based on the slowly varying approximation, they used ANSYS 7.1 software to check the micro-deformation of the optical reference cavity under different placement and support conditions. At the same time, they investigated the factors that affect the accuracy of these simulations. In 2008, Webster et al. [30] proposed the design of an optical reference cavity that is insensitive to force. The design is based on a four-point symmetrical extrusion support mechanism with a cylindrical cavity. The design was improved by finite element analysis, and the vibration sensitivity reached 2.5 × 10^−11^/g (g = 9.81 m/s^2^). In 2009, J Millo et al. [25] improved the design of horizontal and vertical optical reference cavities by using Finite element software. They studied the influence of modifying the vertical constraint in the numerical model on the vibration sensitivity of the cavity.

In 2011, D.R. Leibrandt et al. [31] designed an optical spherical reference cavity, which adopts spherical symmetry and adjustment of installation angle to reduce the influence of three-dimensional vibration, as shown in Figure 2. The vibration sensitivity of this new spherical cavity on three orthogonal axes was 10^−10^/g after outdoor environment evaluation.

In 2012, Y. Jiang [32] developed a finite element model without sliding support points through finite element analysis software and optimized the vertical, horizontal, and annular cavity configuration and support technology. In the same year, B. Argence et al. [33] designed a cylindrical optical reference cavity with a length of 10 cm for space applications, which contained a back plate that could withstand multiple accelerated vibrations in the cavity. The experimental results show that the sensitivity of orthogonal vibration is less than 1 × 10^−10^/g. In 2013, D.R. Leibrandt et al. [34] advanced the design of optical spherical cavities by integrating active real-time compensation for vibration, rotation, and temperature. The experimental results showed that the vibration sensitivity of the system was 10^−12^/g. In 2014, Q. Chen et al. [35] used finite element analysis to simulate the vibration sensitivity of the movable optical reference cavity and achieved the best sensitivity of 10^−11^/g order of magnitude. In 2016, Wu et al. [36] used the finite element method to improve the vibration sensitivity of a 10 cm optical reference cavity and used it to realize a laser with a line width of 0.26 Hz, and achieved remarkable frequency stability of 1 × 10^−15^ in the time scale of 100–4000 s. In the same year, D. Świerad et al. [37] developed a mid-plane vertical cylindrical optical reference cavity for space applications and achieved 7.9 × 10^−16^ laser fractional frequency instability in an average period of 300 ms. In 2018, B. Tao et al. [38] carried out finite element numerical simulation on the vibration sensitivity of a multi-support horizontal optical reference cavity, which was designed to withstand an impact acceleration of 100 g, as shown in Figure 3, and achieved a vibration sensitivity of less than 6 × 10^−11^/g.

In 2019, Lee et al. [39] used the finite element method to analyze the vibration sensitivity of the cavity by changing the support position and area under the fixed (completely constrained) and sliding (only vertically constrained) support conditions. They determined the vibration-insensitive support positions independent of the support area and verified these positions through experimental comparison. In 2020, Xu et al. [40] developed a numerical simulation model of the optical reference cavity and its supporting system using the finite element method, as shown in Figure 4. The model confirms the theoretical prediction of vibration sensitivity caused by elastic deformation in the vertical optical reference cavity.

In 2023, Jiao et al. [41] designed a multi-channel movable spherical optical reference cavity, as shown in Figure 5, whose mass is about half that of a cubic optical reference cavity of similar size. The vibration sensitivity of the cavity was calculated theoretically, and its triaxial acceleration-induced vibration sensitivity was determined to be between 5.1 × 10^−10^ and 8.4 × 10^−10^/g according to experience.

In order to reduce the influence of vibration, the researchers implemented active and passive vibration damping strategies, and greatly enhanced the stability of the optical resonator by optimizing its geometry, supporting frame, and positioning. Many teams use finite element analysis to simulate the micro-deformation of optical resonators in different vibration scenarios, evaluate the influence of different shapes and installation techniques on stability, and then innovate a more flexible resonator design. These studies show that the influence of vibration on the stability of optical resonators can be significantly reduced and the laser frequency stability can be enhanced through structural optimization. Figure 6 below shows a plot of stability reached by different cavities versus years. This provides a crucial insight for minimizing the influence of mechanical vibration on the frequency noise of optical resonators and promoting the development of highly stable optical resonators.

### 3.2. Thermal Noise Control

In order to reduce thermal noise, researchers have done a lot of work. In 1998, Y. Levin [13] introduced a calculation method of thermal noise (TN) based on the wave dissipation theorem (FDT), providing a foundational understanding of thermal noise dynamics. In 2004, Numata et al. [20] expounded the influence of Brownian noise on the length stability of reference cavity according to the displacement spectral density of reference cavity and introduced a formula to calculate the thermal noise of cylindrical superconducting material spacers, reflective substrates, and coatings, establishing a crucial link between material properties and thermal noise performance. In 2007, Tao [42] used finite element software to simulate and analyze the thermal noise of various components of the resonator and found that the thermal noise of the mirror has the greatest influence on the length stability of the whole resonator, followed by the thermal noise of the mirror coating, and the thermal noise of the resonator has the least influence.

In the materials with low coefficient of thermal expansion (CTE), the influence of ambient temperature fluctuation is significantly reduced, so that the optical resonator made of such materials (such as ULE, glass ceramic Zerodur, and monocrystalline silicon) can effectively reduce the influence of thermal noise. In contrast, the CTE of FS is obviously higher than that of ULE, so it is not suitable as an isolation material. However, as a substrate, it can reduce the thermal noise limit of the cavity by 2–3 times compared with ULE. In 2009, Millo et al. [25] studied an F-P cavity combining ULE glass and fused silicon mirror, revealing better thermal noise resistance compared with the cavity made of ULE glass only. In 2010, Thomas Legero et al. [26] studied a low thermal noise F-P cavity, which was composed of a low thermal expansion (LTE) glass lining and a fused quartz mirror. Compared with the all-LTE glass cavity, the zero-crossing temperature was reduced by about 20K. In addition, the finite element simulation and CTE measurement show that the LTE ring, which is in optical contact with the back of the FS mirror, can adjust the zero-crossing temperature in the range of 30K, as shown in Figure 7.

In 2013, Zhang et al. [27] introduced a method to enhance the thermal stability of optical resonators through finite element analysis. The zero-crossing temperature difference of thermal expansion coefficients of various optical resonator models is generated by this analysis, and their relationship with the thickness and diameter of the ULE ring on the surface of the cavity mirror is examined. The goal is to determine the optimal zero-crossing temperature difference of the thermal expansion coefficient and provide information for the design of the optical resonator. Their research underscored the importance of finite element analysis in optimizing the thermal expansion characteristics of cavity mirrors, thereby facilitating the design of more effective optical resonators. In 2021, Xu et al. [43] conducted a comprehensive theoretical analysis and finite element simulation of thermal noise in cubic optical cavities. They developed a theoretical estimation formula for thermal noise of cubic gaskets with any edge size and studied the influence of compressive force on thermal noise in cubic optical cavities for the first time. Their work expanded the theoretical framework by providing a comprehensive analysis of thermal noise in cubic optical cavities, addressing the complexities of compressive forces on thermal performance. The geometric model of finite element analysis is shown in Figure 8.

In 2022, Jiao et al. [44] studied the thermal noise of a cylindrical catenary supported by four gaskets. They use theoretical estimation and finite element simulation to evaluate their influence on thermal noise. Their simulation showed that the displacement noise caused by the support pad was four times higher than that caused by the substrate and GaAs/Al-GaAs coating for a 400 mm long cylindrical structure composed of ultra-low-expansion gasket and molten quartz substrate. This shows that in some materials and design schemes, four support pads are the main sources of thermal noise. Their study explored the thermal noise dynamics in cylindrical structures, revealing the substantial impact of support pad design on noise characteristics. The geometric model used in this study is shown in Figure 9.

Through the development of sophisticated calculation models, careful material selection, and structural optimization, researchers have made significant strides in minimizing thermal noise, thereby enhancing the frequency stability of reference resonators. This progress is vital for the advancement of high-performance optical instruments and precision measurement technologies, underscoring the interconnected nature of theoretical advancements and practical applications in the field.

### 3.3. Temperature Fluctuation Resistance

The influence of the environment on the stability of ultra-stable optical resonators includes not only mechanical vibration but also the stability decline caused by environmental temperature fluctuation. It is very important to study the temperature response behavior of ultra-stable optical reference cavities for their development, and it is the basic design principle to create ultra-stable optical reference cavities that are insensitive to temperature. The thermal design challenges of space-limited ultra-stable optical reference cavity mainly include weakening heat transfer and temperature control [45].

#### 3.3.1. Weaken the Heat Transfer

The reference cavity, which is typically located in a vacuum chamber, only experiences heat from the internal laser, so the external temperature change becomes the main concern of the temperature change in the cavity. Therefore, it is very important to minimize external heat transfer. In 2015, J. Sanjuan et al. [29] developed an optical resonator encapsulated in a thermal shield through numerical simulation and model analysis, which effectively damped external temperature fluctuations. Meanwhile, S. Häfner [46] of PTB in Germany designed a 48 cm ultra-stable cavity and achieved a zero expansion temperature of 0.28 °C. In order to counteract the temperature gradient caused by the cavity length, the multi-point temperature control technology is used to strengthen the first insulation layer, and the third polished insulation layer further reduces the influence of external temperature change, as shown in Figure 10. Room temperature shifts of 0.5 K led to only 500 μK temperature fluctuations within the controlled layer, and a 50 mK departure from the zero expansion point kept cavity temperature fluctuations at ±120 μK. By refining the temperature control system and structural support, the 48 cm cavity’s frequency stability was improved to 8 × 10^−17^, approaching the thermal noise limit.

In order to reduce the instability of cavity length caused by temperature, the reference cavity is usually accommodated in a vacuum chamber supported by an ion pump. A plurality of thermal shielding layers is placed between the vacuum chamber and the reference chamber to mitigate the influence of external temperature fluctuation. In 2019, J. Sanjuan et al. [47] adopted five thermal shielding layers to minimize the influence of ambient temperature on the reference cavity, and realized an ultra-stable laser with significant long-term stability, as shown in Figure 11.

#### 3.3.2. Temperature Control

Due to the inevitable external temperature fluctuation and the extreme sensitivity of the spatial ultra-stable optical reference cavity to this change, accurate environmental control of the reference cavity is essential while also minimizing external heat transfer. In 2012, Guo Haifeng [48] implemented a one-way temperature control system for the vacuum chamber, using polyimide electrothermal film for heating, and then put the vacuum chamber in a thermostat for further temperature adjustment, as shown in Figure 12.

In 2015, Dai et al. [49] analyzed the material properties of the ultra-stable cavity and the mirror and adjusted the TiO_2_ doping level of ULE, which made the zero expansion temperature closer to room temperature, which was helpful for accurate system temperature control.

In 2017, Wang et al. [50] developed a vacuum temperature control system for an optical Fabry–Perot cavity made of ultra-low expansion glass materials. As shown in Figure 13, the temperature of the cavity was accurately maintained between 10 and 40 °C through the dual temperature control mechanism. The system exhibited a temperature variation of about ±0.004 °C over 24 h, with the cavity’s zero expansion temperature being 29.286 ± 0.057 °C.

In 2020, Guo [51] adopted a thermoelectric cooler and temperature feedback system to adjust the temperature of the vacuum chamber to ensure the constant temperature environment of the F-P cavity so as to further stabilize its length, as shown in Figure 14.

In 2023, Zhao et al. [52] proposed a thermal compensation “collar”, which was in optical contact with the resonant cavity (Figure 15). This is to solve the thermal mismatch between FS substrate and ULE gasket, and also to check the flexible adjustment of zero thermal expansion temperature. Compared with the full ULE cavity, this method allows the temperature difference adjustment range from −10 K to 23 K. It is helpful to design the zero thermal expansion temperature of the optical reference cavity slightly higher than room temperature, thus simplifying the temperature control challenge.

These studies enhance the stability of ultra-stable optical resonators to temperature changes and enable them to maintain high-precision performance in a wider temperature range. The goals are to reduce the external temperature fluctuation by making complex heat insulation and temperature control systems and to maintain the thermal stability in the cavity by adopting accurate temperature control technology, thus expanding its application scope.

## 4. Typical Applications

Improving the stability of optical resonators is the result of theoretical research and the basis of various application innovations. With the progress of science and technology, the requirements for the accuracy and stability of optical systems have been improved, which makes optical resonators very important in precision navigation, high-precision measurement, quantum communication, and gravitational wave detection. These applications emphasize the extensive influence and important role of optical resonator stability research in promoting scientific and technological progress.

### 4.1. Laser Gyroscopes

Laser gyroscopes are widely employed for inertial navigation [53,54], and also in geophysics for precise metrology [55,56,57,58,59]. In the field of navigation, the laser gyroscopes, which serve as the core device of the navigation system in weapons and carriers, have not been the subject of much recent public information. The laser gyroscope GG1389, developed by Honeywell, has been reported to demonstrate the highest accuracy and zero bias stability of 1.5 × 10^−4^/h, as illustrated in Figure 16. The GG1389 has a triangular cavity configuration with a cavity path length of 68 cm [60]. In terms of large laser gyroscopes for high-precision geophysical measurements, representative ones include GINGERINO, whose active ring laser upper limit noise approaches 2 × 10^−15^ rad/s at an integration time of about 2 × 10^5^ s. The GINGERINO project is illustrated in Figure 17. Such high-precision measurements provide a solid foundation for the verification of general relativity, for the measurement of Earth’s velocity, and for the study of quantum physics for tiny effects.

The improvement of measurement accuracy in laser gyroscopes is a significant research topic at present. The accuracy of laser gyroscopes depends on the accuracy and stability of beat frequency and scale factor (*λL*/4*A*). Enhancing the accuracy of laser gyroscopes necessitates the stabilization of the optical path length (*L*) within the ring cavity and the minimization of fluctuations in laser output power. This requires optimizing the design and material selection of optical resonators and improving thermal and mechanical stability [62]. Using materials with low thermal expansion coefficients and high-precision temperature control systems can reduce the influence of temperature fluctuation on optical path length, thus enhancing the stability of gyroscopes. In addition, it is very important to ensure the precise geometry of the resonator through advanced processing technology for stabilizing the optical path length [63].

In addition, it is very important to improve the vibration resistance of gyroscopes. Vibration will change the cavity structure of laser gyroscopes, resulting in the deformation of four internal reflection points. These deformations can be used to evaluate the influence of vibration on laser gyroscopes. By establishing the finite element model, the deformation of the optical resonator can be quantitatively analyzed, and the influence of resonator size, aperture, and rotor installation mode on the deformation of the reflection point can be studied. Based on these analyses, the optimization scheme of the vibration mechanism can be put forward to improve the connection strength, reduce the internal stress, and alleviate the temperature effect, thus further improving the design of laser gyroscopes.

The closed-loop feedback control system is used to monitor the real-time changes in the cavity length of the ring, and these changes are compensated by adjusting the laser frequency or changing the length of the resonator, which ensures that the working longitudinal mode frequency remains within the gain distribution.

These optimization measures will significantly improve the measurement accuracy and stability of laser gyro, thus improving the overall performance.

### 4.2. High-Precision Measurement Technology

High-precision measurement techniques, including laser interferometer [64,65,66], cavity optomechanics [67,68,69], and atomic clock [70,71], rely on extremely stable light sources to ensure the accuracy and reliability of measurement. The optical resonator is the core component of a laser, and its stability is very important to maintain the stability of laser output frequency, thus affecting the performance of the whole measurement system.

For example, a laser interferometer uses the interference of multiple laser beams to accurately measure physical parameters such as length, displacement, and angle. The stability of the optical resonator affects the phase stability of the laser beam in the interferometer, thus affecting the measurement accuracy. Similarly, in atomic clocks, especially optical atomic clocks that use atomic transition frequency as a time standard, stability, and accuracy are very important for applications such as global positioning systems (GPSs), deep space navigation, and basic physics research. An optical resonator stabilizes the laser frequency used to excite specific atomic transitions, which determines the time standard stability of atomic clocks.

By using materials with low thermal expansion coefficients, complex temperature control, and vibration isolation, the progress in the stability of optical resonators has significantly enhanced the high-precision measurement technology. However, further improvement of measurement accuracy needs to solve technical challenges, such as nonlinear effects and losses caused by material absorption and scattering in the optical resonator.

### 4.3. Gravitational Wave Detection

Gravitational wave (GW) detection is a frontier field of modern physics, aiming at directly observing the space–time distortion caused by celestial motion [72,73]. Because the gravitational wave signal is extremely weak, which leads to small spatial distance change, highly stable optical equipment is very important for its measurement [74].

A GW detector like LIGO (Laser Interferometer GW Observatory) is essentially a large laser interferometer, and its core is a highly stable optical resonator. LIGO has implemented a temperature control system with micro-Kelvin noise levels to mitigate the impact of temperature fluctuations on measurement noise. The system helps to maintain a stable temperature and prevent any interference caused by temperature changes. Additionally, the device is enclosed in vacuum chambers to further minimize external interference [75]. These cavities generate very stable laser beams to measure the tiny length changes in the interferometer arm, to detect gravitational waves passing through the earth. Additionally, in the space-based GW detector LISA and the TianQin-1, temperature error suppression techniques such as ultra-low expansion glass (ULE) and Zerodur substrates have been effectively validated [76,77].

For the necessary sensitivity in gravitational wave detection, optical resonators need high-frequency stability and low noise. Stability problems caused by temperature fluctuation, mechanical vibration, or external interference may blur or imitate gravitational waves, leading to wrong detection. Therefore, the detector design combines complex temperature control, vibration isolation technology, and vacuum conditions to ensure the ultra-high stability of the optical resonator [78,79,80].

Technological progress has greatly enhanced the stability and sensitivity of the optical resonator in gravitational wave detectors, and realized the direct detection of gravitational wave events for the first time, thus creating a new era of gravitational wave astronomy.

However, with the continuous improvement of detection sensitivity, the demand for the stability of optical resonators is also increasing. Future gravitational wave detection will depend on further technological innovation to detect weaker gravitational wave events and promote gravitational wave physics and cosmology research.

## 5. Conclusions and Future Directions

By discussing the related principles, key technical advancements, and typical applications of optical resonator stability, the significance and extensive influence of improving the stability of optical resonators are demonstrated. Significant progress has been made by adopting effective vibration isolation, effective thermal noise suppression, and advanced temperature control technology. These advances have solved the stability problems caused by mechanical vibration and temperature changes and laid a solid foundation for the application of optical resonators. In addition, enhancing the stability of the optical resonator greatly promotes high-precision measurement technology and quantum communication, which provides higher accuracy and reliability for the laser gyro and provides necessary technical support for gravitational wave detection. These successful applications emphasize the great achievements in the study of optical resonator stability and its great potential for future scientific and technological efforts. With the progress of science and technology and the deepening of research, the demand for the stability of optical resonators will gradually increase. Future research will span different fields, including material science, mechanical engineering, optical design, and control science. Interdisciplinary research and technological innovation aimed at achieving higher stability of optical resonators will meet the needs of precision measurement and quantum communication, and put forward key areas for future stability research of optical resonators:(1)Application and exploration of new materials

The progress of materials science is important for finding and using new materials with low expansion and high stability to improve the stability of optical resonators. The research on new materials such as optical-grade glass, crystal, and composite materials is the key to improving thermal noise treatment and thermal expansion coefficient, which directly affect the temperature and mechanical stability of optical resonators. Future work should focus on the selection and improvement of this material to minimize its sensitivity to temperature changes and mechanical vibration.

(2)Advanced vibration isolation technology

Although a series of vibration isolation technologies have been applied, including active and passive methods, the isolation of very-low-frequency vibration is still a major challenge. It is very important to study advanced vibration isolation technology, such as reducing mechanical vibration based on nonlinear dynamics or superconducting magnetic levitation, to improve the stability of optical resonators.

(3)Intelligent temperature control system

The development of an intelligent and high-precision temperature control system represents a research focus, and its goal is to solve the temperature fluctuation that affects the stability of optical resonators. This includes using complex sensors and feedback control algorithms to carefully monitor the ambient temperature and adjust the optical resonator in real time, thus minimizing temperature fluctuations.

(4)Design of multifunctional integrated optical resonator

In the future, the design of optical resonators will exceed the current stability standards and enhance its versatility. This will include combining self-correction and self-adjustment capabilities to make the resonator adapt to different environments and applications. This will require the integration of optical, electronic, and mechanical designs to develop an optical resonator that can automatically compensate its environment and output multiple frequencies.

(5)Deepening of theoretical model and numerical simulation.

The enhancement of computing power and the progress of numerical simulation will help to create more accurate and complex optical resonator models, thus promoting a deeper understanding of internal physical processes and external environmental factors. The progress of theoretical modeling and simulation research will help to generate information related to experimental design, thus reducing the cost of trial and error and accelerating the development of optical resonator technology.

In essence, improving the stability of optical resonators is very important for theoretical exploration and contemporary scientific and technological progress. It is expected that the continuous optimization of optical resonator design and performance will contribute to its future application in a series of fields.

## Figures and Tables

**Figure 1 sensors-24-06473-f001:**
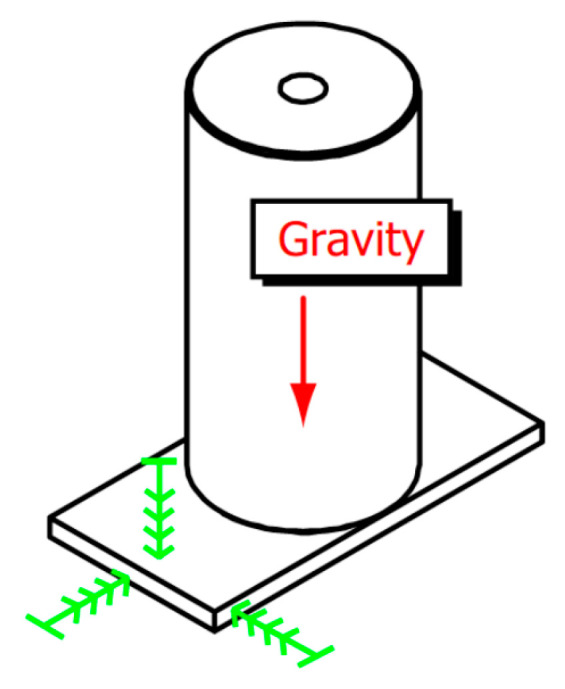
A vertically placed cylindrical cavity, with gravity acting on the cavity, supported at the bottom by a padded plate (the straight lines with multiple arrows indicate support) [6].

**Figure 2 sensors-24-06473-f002:**
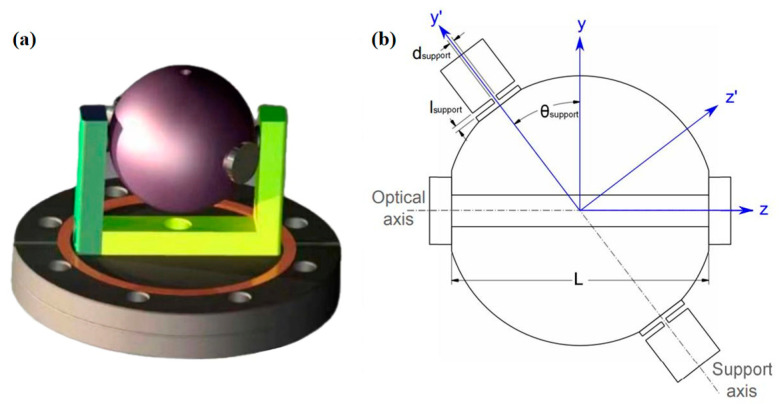
(**a**) A CAD depiction illustrates a spherical cavity mounted at a compression-insensitive angle, utilizing Viton O-ring contacts, designated as the experimental design. (**b**) A sectional illustration of a spherical cavity, positioned at a compression-insensitive angle, features cylindrical ULE contacts [31].

**Figure 3 sensors-24-06473-f003:**
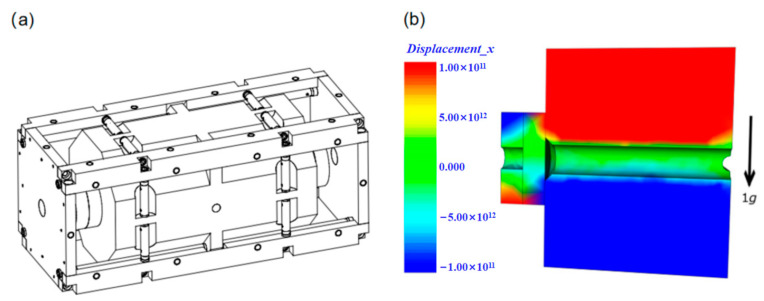
(**a**) Diagram of the cavity and its support structure. (**b**) FEA simulation output for cavity axial displacement under 1 g acceleration transverse to the cavity [38].

**Figure 4 sensors-24-06473-f004:**
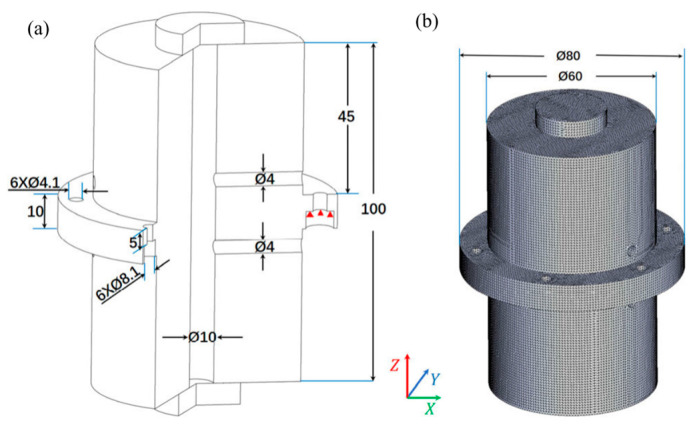
Geometric dimensions of the optical reference cavity and simulation model. (**a**) Cross-sectional drawing and geometric dimensions of the optical reference cavity with the cavity spacer and the mirrors, the central ring with fixation holes, and the two through venting holes, and red triangles indicate constraints. (**b**) The simulation model with meshed elements [40].

**Figure 5 sensors-24-06473-f005:**
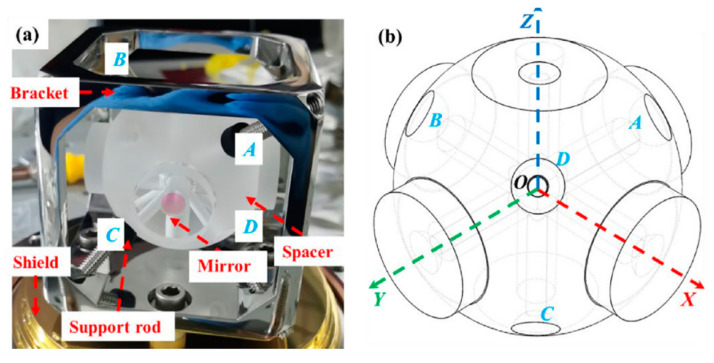
Geometric representation of a four-point supported spherical optical reference cavity: (**a**) Mounted on a support bracket. (**b**) The cavity itself [41].

**Figure 6 sensors-24-06473-f006:**
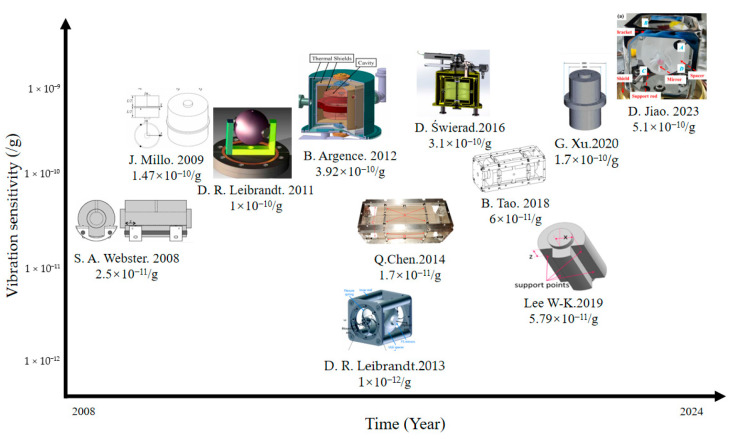
The relationship between the stability of different cavities and the year.

**Figure 7 sensors-24-06473-f007:**
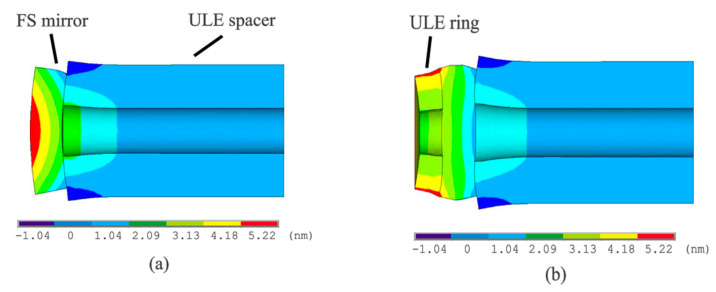
FEM analyses of the elastic deformation of the cavity following a 1K temperature shift: (**a**) FS mirror optically bonded to a ULE spacer. (**b**) An added ULE ring at the FS mirror’s rear to mitigate its axial warping [26].

**Figure 8 sensors-24-06473-f008:**
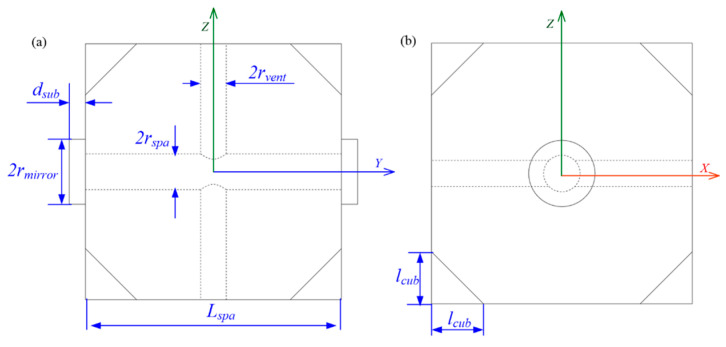
Diagram of the cubic optical resonator for analysis and FEA: (**a**) side view; (**b**) front view [43].

**Figure 9 sensors-24-06473-f009:**
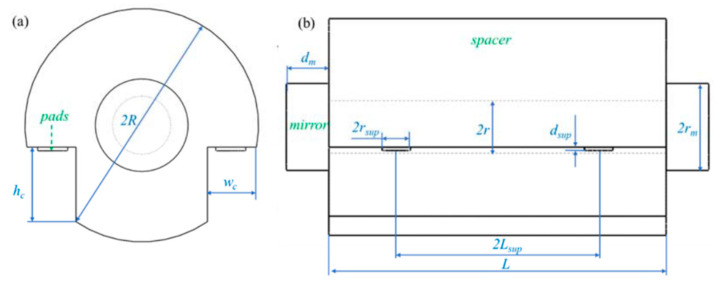
Geometric model of cylindrical USC with four elastic supports: (**a**) front view; (**b**) side view [44].

**Figure 10 sensors-24-06473-f010:**
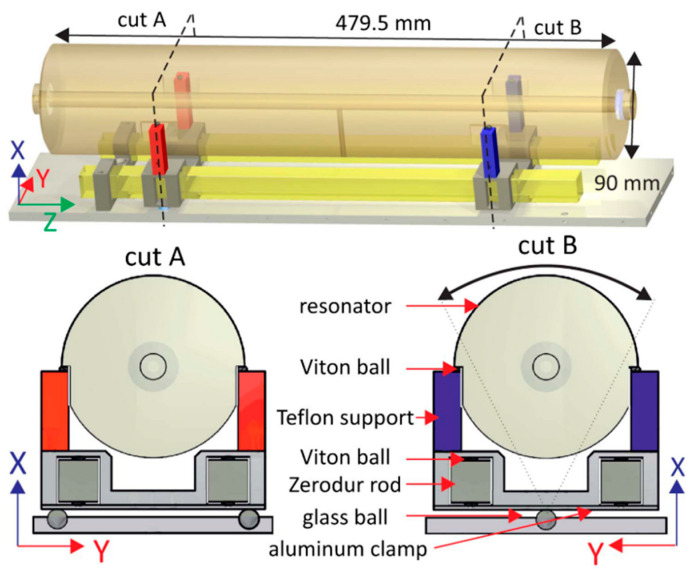
Resonator mounting design illustrated by cross-sections of fixed (A) and rotatable (B) supports [46].

**Figure 11 sensors-24-06473-f011:**
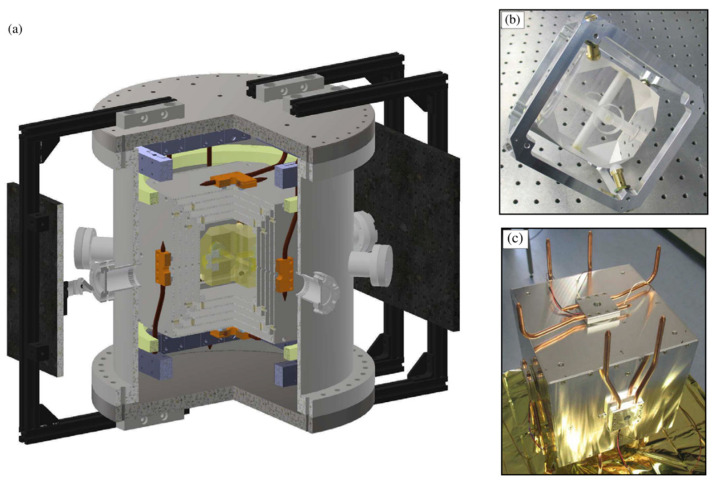
(**a**) Diagram of the setup. (**b**) The 8.7 cm ULE cavity with a tetrahedral mount. (**c**) External thermal shield layer featuring Peltier elements (concealed by adapters) and heat pipes [47].

**Figure 12 sensors-24-06473-f012:**
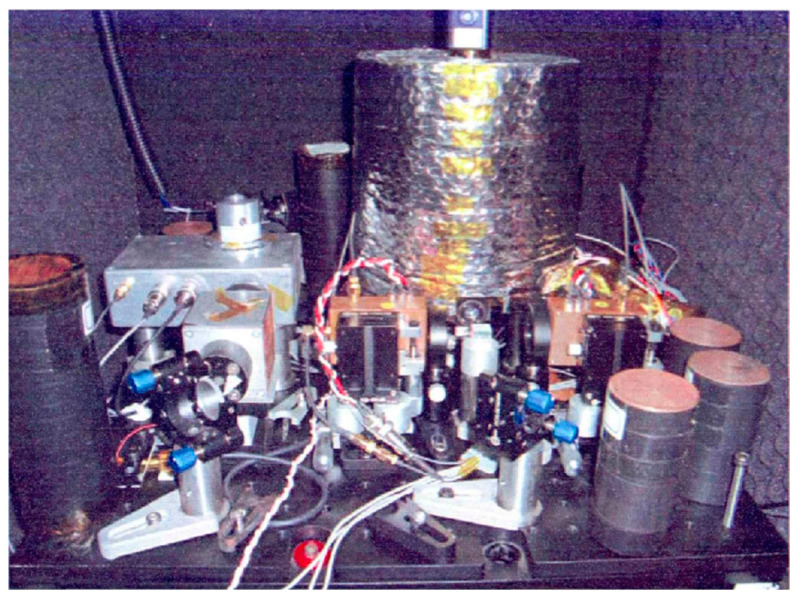
Temperature control system of vacuum chamber [48].

**Figure 13 sensors-24-06473-f013:**
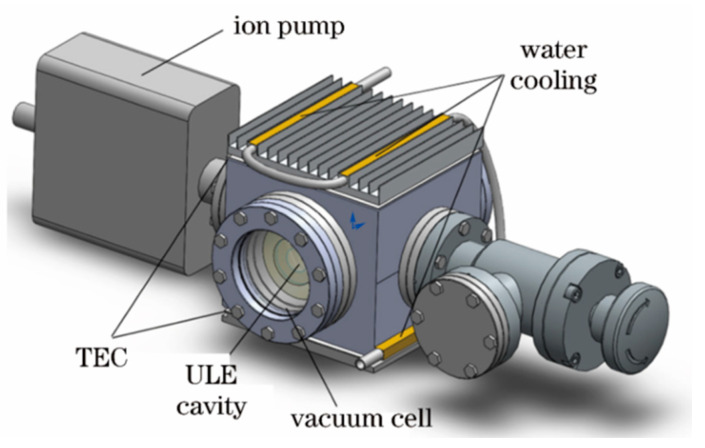
Schematic diagram of the mechanical mechanism of high-thermal-stability cavity vacuum system [50].

**Figure 14 sensors-24-06473-f014:**
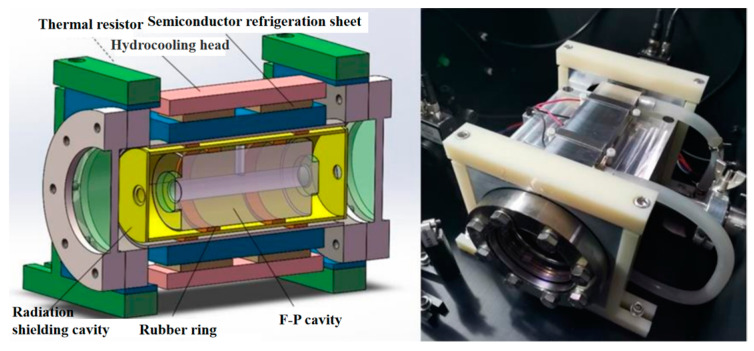
Design drawing (**left**) and physical drawing (**right**) of the temperature control system [51].

**Figure 15 sensors-24-06473-f015:**
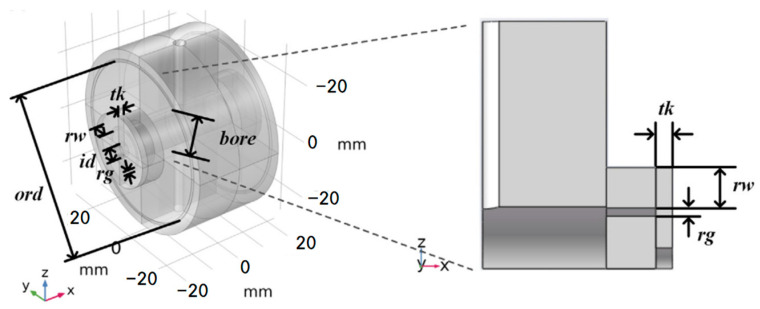
3D model and of the cross-section of the “lantern ring” structure cavity [52].

**Figure 16 sensors-24-06473-f016:**
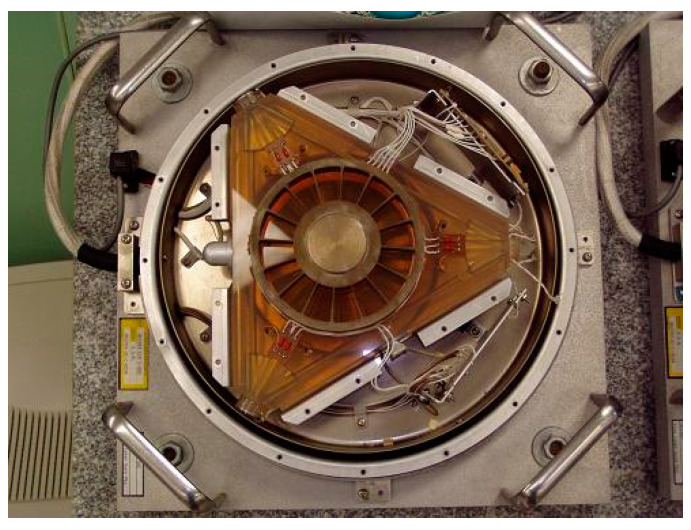
The GG1389 ring laser gyroscope.

**Figure 17 sensors-24-06473-f017:**
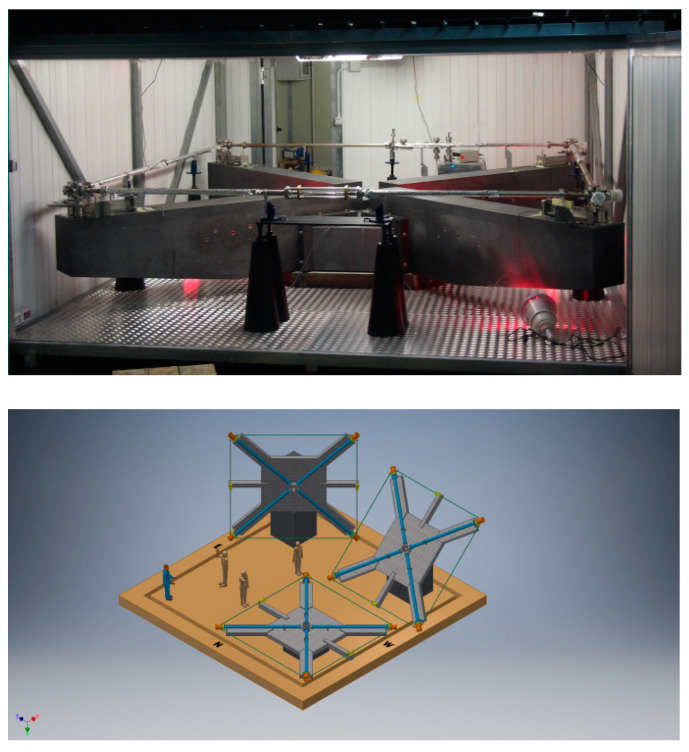
Schematic of GINGERINO and The GINGER project [61].

## Data Availability

The original data presented in the study are openly available in reference.

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
