# Peer review of "Advancements in Optical Resonator Stability: Principles, Technologies, and Applications"

_sensors, 2024, doi:10.3390/s24196473_

Round 1

Reviewer 1 Report (New Reviewer)

Comments and Suggestions for Authors

The paper covers a significant and complex topic in optical resonator stability, offering a broad overview of current advancements and future directions. This paper can serve as a valuable resource for researchers and practitioners in the field.

After minor revisions, this paper can be considered for acceptance with the following changes needed:

1.       Ensure consistent use of terminology throughout the paper. For example, if "optical resonator" is used, avoid switching between similar terms like "optical cavity" without clear distinction.

 2.       The introduction does not highlight the contributions of this paper.

 3.       The introduction could provide a clearer outline of the paper’s structure, briefly summarizing what each section will cover. This helps guide readers through the paper.

 4.       In introduction, Change "On the flip side" to "Conversely" for a more formal tone.

 5.       Ensure that the formatting of equations, figures, and references is consistent throughout the paper. For example, in-text citations should have spaces before brackets (e.g., "axis [1]" instead of "axis[1].").

 6.       when discussing thermal noise control methods, provide more detail on how these methods have evolved and their impact on current technology.

 7.       The clarity of all the pictures in the first manuscript is poor, I hope to improve the clarity of each picture in the final manuscript.

 8.       In "By expounding the related principles, key technical progress and typical applications of optical resonator stability," it may be clearer to write "By discussing the related principles, key technical advancements, and typical applications of optical resonator stability,..."

Author Response

  1. Ensure consistent use of terminology throughout the paper. For example, if "optical resonator" is used, avoid switching between similar terms like "optical cavity" without clear distinction.

Response:Thank you very much for providing such valuable suggestions. We have made the necessary revisions.

  1. The introduction does not highlight the contributions of this paper.

Response:Thank you for your suggestion, which has been supplemented and explained in Section 1.

  1. The introduction could provide a clearer outline of the paper’s structure, briefly summarizing what each section will cover. This helps guide readers through the paper.

Response:Thank you very much for the valuable suggestions you provided. We have made revisions in the introduction section.

  1. In introduction, Change "On the flip side" to "Conversely" for a more formal tone.

Response:Thank you very much for your valuable suggestions. We have made the revisions.

  1. Ensure that the formatting of equations, figures, and references is consistent throughout the paper. For example, in-text citations should have spaces before brackets (e.g., "axis [1]" instead of "axis[1].").

Response:Thank you very much for your valuable suggestions. We have made the revisions.

  1. when discussing thermal noise control methods, provide more detail on how these methods have evolved and their impact on current technology.

Response:Thank you very much for making such valuable suggestions, and we have already supplemented this aspect, as shown in Section 3.2.

  1. The clarity of all the pictures in the first manuscript is poor, I hope to improve the clarity of each picture in the final manuscript.

Response:Thank you for your suggestion. The pictures in the paper have been updated to make them clearer.

  1. In "By expounding the related principles, key technical progress and typical applications of optical resonator stability," it may be clearer to write "By discussing the related principles, key technical advancements, and typical applications of optical resonator stability,..."

Response:Thank you very much for your valuable suggestions. We have made the revisions.

Reviewer 2 Report (New Reviewer)

Comments and Suggestions for Authors

The paper presents a thorough overview of achievements in optical resonator stability, covering both theoretical and practical aspects. This comprehensive approach is valuable for readers seeking to understand the state-of-the-art in this field. Upon conducting an exhaustive review of the manuscript, I think this manuscript could be basically accepted after addressing the following matters.

1.       The English language could be improved. There are a few paragraphs where sentences can be shorted or refined.

2.       It is suggested to highlight the paper's novelty and impact both in the introduction and conclusion, such as new insights into optical resonator stability or the proposal of innovative techniques, to emphasize the paper's significance.

3.       Maintain consistent technical terminology throughout the paper.

4.       It is recommended to use a table to summarize the relevant technologies, making it easier for readers to quickly understand the advantages, disadvantages, and comparisons between different techniques.

5.       It is suggested to include brief formulas or diagrams in the theoretical introduction of section 2.1 for better illustration. 

6.       It seems many figures are blurring. The resolution is insufficient. Please double check.

Comments on the Quality of English Language

Moderate editing of English language required.

Author Response

1.The English language could be improved. There are a few paragraphs where sentences can be shorted or refined.

Response:Thank you for your detailed suggestions. We have addressed the issues you pointed out and shortened and refined several sentences.

2.It is suggested to highlight the paper's novelty and impact both in the introduction and conclusion, such as new insights into optical resonator stability or the proposal of innovative techniques, to emphasize the paper's significance.

Response:Thank you for your suggestion, which has been supplemented and explained in Section 1.

3.Maintain consistent technical terminology throughout the paper.

Response:Thank you very much for providing such valuable suggestions. We have made the necessary revisions.

4.It is recommended to use a table to summarize the relevant technologies, making it easier for readers to quickly understand the advantages, disadvantages, and comparisons between different techniques.

Response:Thank you very much for your suggestion. We will summarize some of the research findings in Figure 6. In other sections, it has been challenging to encapsulate the outcomes of various research types in graphs and tables, hence we have not been able to visualize them in a tabulated form. We hope you can understand this situation.

5.It is suggested to include brief formulas or diagrams in the theoretical introduction of section 2.1 for better illustration.

Response:Thank you for your valuable comments, which we have taken on board by adding schematic diagrams, and formulas for illustrating the problem in Section 2.1.

6.It seems many figures are blurring. The resolution is insufficient. Please double check.

Response:Thank you for your suggestion. The pictures in the paper have been updated to make them clearer.

7.Moderate editing of English language required.

Response:Thank you for your suggestion regarding the need for moderate editing of the English language. We will carefully review and revise the language in the paper to enhance its clarity and fluency. We appreciate your valuable feedback!

This manuscript is a resubmission of an earlier submission. The following is a list of the peer review reports and author responses from that submission.

Round 1

Reviewer 1 Report

Comments and Suggestions for Authors

This manuscript is not suitable for publication and must be rejected since a significant part of it (maybe the whole manuscript) was 100% written by ChatGPT or similar AI generative software.

Indeed, I scanned the text of Abstract, Sections 1, 4, 5 by the website https://gptzero.me and I got the same response: 100% probability that these sections were entirely written by AI.

Below I list some clues that aroused my suspicion about the non-human origin of this manuscript.

---

For good visibility, all figures should be enlarged to at least 75% of the width of the text.

p.2, l.95

"[((U-1)Hz)]"

This expression is meaningless.

p.5, l.202

"In 2006 Chen et al. [7]"

This citation is wrong, since Ref. [7] is by Nazarova et al.

p.5, l.206-207

"In 2009, J. Millo and collaborators [29] utilized ANSYS software"

Ref. [29] is wrongly cited, ref. [24] should be cited instead.

ANSYS software is not mentioned at all in Ref. [24].

p.5, l.209-210

"Also in the same year, L. Ma and colleagues..."

No Ref. is cited here.

p.5, l.211

"In 2011, Webster and colleagues [30]"

Ref. [30] was published in 2008.

p.5, l.213

"cubic cavity"

It was a cylindrical cavity.

p.5, l.213-214

"This design was subsequently refined..."

No Ref. is cited here.

p.5, l.215

"In the corresponding year, D. R. Leibrandt and colleagues [31]..."

The year 2011 should be explicitly mentioned here.

p.6, l.230

"1 times 10-10 per g"

The correct writing of this expression is "1 x 10-10 / g".

Several similar instances of this incorrect writing can be found in the manuscript.

p.6, l.244

"superior to 6 times 10-11/g"

It is just the opposite: "lower than 6 x 10-11 / g"

Fig. 3

It does not belong to Ref. [40] and it does not depict a vertical cavity at all.

p.8, l.311-214

"In 2021, G. Xu and team [43] conducted a comprehensive theoretical analysis and finite element simulation on Boulanger noise within cubic optical cavities. They developed a theoretical estimation formula for Boulanger noise"

Ref. [43] does not mention Boulanger noise at all.

p.9, l.319-320

"cylindrical ultrafine catenaries"

This expression is completely meaningless in this context.

Fig. 10

It does not belong to Ref. [48].

p.11, l.378

"In 2015, Dai. X and colleagues [9,49]"

Citation of Ref. 9 here is meaningless.

Fig. 12

It does not belong to Ref. [51].

Sec. 4

Many of the Refs. cited in this section [53-60] are completely off topic.

Sec. 4.3

To my knowledge, high-stability optical cavities are not relevant for quantum communication.

References

[50] The correct journal name is "Acta Optica Sinica"

[54] The journal name is repeated twice

Reviewer 2 Report

Comments and Suggestions for Authors

The paper presents a review on optical cavity stability. The thermal noise, vibration and temperature characteristics of reference cavities have been discussed. Moreover, advancements in mechanical vibration isolations, thermal noise control and resistance to temperature fluctuations have been investigated. Finally, few applications of optical cavities have been discussed.

Upon a thorough examination of the manuscript, I find it to be a valuable survey for readers seeking an introduction to the field. I am pleased to recommend its publication in Sensors; however, prior to formal acceptance, I have a few suggestions for the authors' consideration:

_ Line 61 I recommend the following paper to cite: An introduction to Pound–Drever–Hall laser frequency stabilization Am. J. Phys. 69, 79–87 (2001) https://doi.org/10.1119/1.1286663

_ Line 77-80: the sentence is not clear to me. I recommend rephrasing the sentence.

_ Line 94: I think it is better to specify which is the high frequency regime.

_ Line 95: It is not clear to me the meaning of the abbreviations U and -l.

_ Line 126: the wrong symbol has been used. The Authors probably meant \phi.

_ Line 131: “where” does not need to be in capital letter.

_ Section 2.3: the Authors says to keep the cavity as short as possible to minimize temperature variations, but a short cavity has a very large linewidth losing its frequency stabilization property. Could the Authors, please, clear this point?

_ Section 3.1: a plot of stability reached by different cavities versus years might improve the readability of the review showing the progress in the field.

_ Line 202, line 208, line 211, line 224: the references in the text do not match the bibliography.

_ Section 3.1: do the materials used to build the support for the mirrors of the cavities play a role in the vibration mechanical noise?

_ Line 423: an explanation of the terms in the equation of the scale factor is needed.

_ Section 4.1: showing the state-of-the-art sensitivity of the gyroscopes might help to understand the limits and how improving in the cavity stability might help for the sensitivity of future gyroscopes.

_ Section 4.4: It might be worth it to introduce few papers about Optomechanics where the mechanical motion at the picometer level is detected using Fabry Perot cavities. With optical cavities you can reach the sensitivity to measure ground state cooling. I provide few papers which the authors might use to expand the discussion in order to strengthen the need of high stable cavities:

_ 10.1103/RevModPhys.86.1391
_ 10.1038/s41586-018-0643-8
_ 10.3390/photonics9020099

Line 502: It is not clear to me how mechanical vibrations or temperature fluctuations might mimic a gravitational wave detection as they have a well-known pattern. A reference might help.

Line 509-510: It might be useful to provide a reference to justify the statement.